# Genetic Analysis for the Flag Leaf Heterosis of a Super-Hybrid Rice WFYT025 Combination Using RNA-Seq

**DOI:** 10.3390/plants12132496

**Published:** 2023-06-29

**Authors:** Qin Cheng, Shiying Huang, Lan Lin, Qi Zhong, Tao Huang, Haohua He, Jianmin Bian

**Affiliations:** 1Key Laboratory of Crop Physiology, Ecology and Genetic Breeding, Ministry of Education, Jiangxi Agricultural University, Nanchang 330045, China; 15797670057@163.com (Q.C.); syhuang1998@163.com (S.H.); 18270706636@163.com (L.L.); 18279658618@163.com (Q.Z.); ht19980528@163.com (T.H.); 2College of Agronomy, Jiangxi Agricultural University, Nanchang 330045, China

**Keywords:** indica, super-hybrid rice, heterosis, grain number, flag leaf, RNA-seq

## Abstract

The photosynthetic capacity of flag leaf plays a key role in grain yield in rice. Nevertheless, there are few studies on the heterosis of the rice flag leaf. Therefore, this study focuses on investigating the genetic basis of heterosis for flag leaf in the indica super hybrid rice combination WFYT025 in China using a high-throughput next-generation RNA-seq strategy. We analyzed the gene expression of flag leaf in different environments and different time periods between WFYT025 and its female parent. After obtaining the gene expression profile of the flag leaf, we further investigated the gene regulatory network. Weighted gene expression network analysis (WGCNA) was used to identify the co-expressed gene sets, and a total of 5000 highly expressed genes were divided into 24 co-expression groups. In CHT025, we found 13 WRKY family transcription factors in SDG_hp_s under the environment of early rice and 16 WRKY family genes in SDG_hp_s of under the environment of middle rice. We found nine identical transcription factors in the two stages. Except for five reported TFs, the other four TFs might play an important role in heterosis for grain number and photosynthesis. Transcription factors such as *WRKY3*, *WRKY68,* and *WRKY77* were found in both environments. To eliminate the influence of the environment, we examined the metabolic pathway with the same SDG_hp_ (SSDG_hp_) in two environments. There were 312 SSDG_hp_s in total. These SSDG_hp_s mainly focused on the phosphorus metallic process, phosphorylation, plasma membrane, etc. These results provide resources for studying heterosis during super hybrid rice flag leaf development.

## 1. Introduction

Heterosis is a phenomenon in which hybrids are superior to their parental lines in economic traits, which is an important strategy for crop breeding. A significant increase in rice grain yield in recent decades has been attributed to the use of heterosis in hybrids. At the genomic level, three hypotheses, namely dominant complementarity, over-dominance [1] and epistasis [2], are often proposed to explain the mechanism of heterosis. Although heterosis is widely used to enhance crop productivity worldwide, its molecular basis remains enigmatic. With the development of molecular marker technology and QTL mapping, geneticists used population genetic analysis in combination with molecular markers for QTL mapping of heterosis. Nevertheless, the partitioning of dominant populations and genetic diversity cannot fundamentally explain the heterosis that occurs between the offspring hybrids and their parents [3].

To uncover heterosis in hybrid rice, researchers used omics methods to find a number of genetic loci related to heterosis [4]. High-throughput RNA sequencing (RNA-seq) for in-depth transcript analysis contributes significantly to a better understanding of heterosis in rice [5]. Researchers analyzed the root transcriptomes of super-hybrid rice cultivar Xieyou 9308 and its parents at tillering and heading stages. They found that transcripts were differentially expressed between WFYT025 and its parents (DGHP) and identified a group of potential candidate transcripts [6]. Scientists used RNA-Seq technology to analyze the transcriptome of two rice hybrids, i.e., Ajay (based on wild-abortive (WA)-cytoplasm) and Rajalaxmi (based on Kalinga-cytoplasm), and their respective parents at the panicle formation (PI) and grain filling (GF) stages. The results showed that a number of important transcription factors (TFs) are encoded by the identified distinct expression genes (DEGs) [7]. E et al. used RNA-seq data from two rice genotypes and their reciprocal hybrids and identified genes that are differentially expressed or allele-specifically expressed in the hybrids and their parents. They identified 45 ASE (allele-specific expression) transcription factors belonging to 17 families [8]. Shao et al. performed a genome-wide analysis of ASE by comparing the reading ratios of SNPs of parental alleles in an elite rice hybrid (Shanyou 63) and its parents [Zhenshan 97 (ZS97) and Minghui 63 (MH63)] using RNA-seq data from seedling shoots, flag leaves, and young panicles of plants grown under four environmental conditions [9]. The results suggest that consistent ASEGs can cause partial to complete dominance effects on the traits they regulate, while directionally shifting ASEGs can cause overdominance.

Increasing food production is a long-term goal of crop breeding to meet the demands of global food security [10]. In rice, the uppermost three leaves, especially the flag leaf, are the main source of carbohydrates supply for grain development [11]. It is reported that the flag leaf produces more than 50% of the carbohydrates for the grains. Leaf size and angle, as the main components of plant architecture, affect photosynthesis in the canopy and crop productivity [12,13]. Rice leaf expansion is largely determined by longitudinal cell division, cell elongation, and cell expansion. Some genes have been shown to be associated with rice leaf phenotype, such as *NARROW LEAF 1* (*NAL1*) [14], *NAL2* and *NAL3* [15], *NAL7* [16], *NARROW AND ROLLED LEAF 1* (*NRL1*) [17], and *ABNORMAL VASCULAR BUNDLES* (*AVB*) [18]. However, there are few studies on the heterosis of the rice flag leaf. Therefore, it is necessary to investigate the heterosis of the rice flag leaf.

In this study, we focused on heterosis in the rice cultivar WFYT025, which is widely grown in China. The length, width, and leaf area of the flag leaf showed great differences between WFYT025 and its female parent. Therefore, we sampled the flag leaf of both WFYT025 and its two parents for high-throughput transcriptome sequencing to investigate genes related to photosynthesis or transpiration and seed development. Gene regulatory network analysis revealed several large sub-networks representing interactions between genes with similar expression profiles, which were termed co-expression modules. Uncovering the function of these transcripts may provide useful information for understanding the molecular mechanism of heterosis. Several major sub-networks were found to represent interactions between genes with similar expression profiles via gene regulatory network analysis. Co-expressed gene sets were identified by weighted gene co-expression network analysis (WGCNA), and a total of 5000 highly expressed genes were divided into 24 co-expression groups. We found nine identical transcription factors in both phases. Except for five reported TFs, the other four TFs might play important roles in grain number and photosynthetic heterosis.

## 2. Results

### 2.1. Phenotypic Analysis of WFYT025 and Its Parents

In this study, we examined flag leaf length, width, leaf area, and 1000 spikelet weight or 1000-grain weight (TGW) of WFYT025 and its two parents at 1 day post-anthesis (DPA) and 10 DPA under different environmental conditions (Table 1). In the early rice environment, WFYT025 exhibited greater leaf length, leaf width, leaf area, and Spikelets weight at 1 DPA compared with WFB and CHT025. However, the leaf length of WFYT025 was longer than that of CHT025 but shorter than that of WFB, and the TGW of WFYT025 was higher than that of CHT025 but lower than that of WFB at 10 DPA. In the middle rice environment, the photosynthetic rate (μm/m^2^/s) of WFB was 12.63, CHT025 32.08, and WFYT025 37.95 at 1 DPA, WFB 30.53, CHT025 36.50 and WFYT025 30.35 at 10 DPA. The transpiration rate (μm/m^2^/s) of WFB was 4.55, CHT025 11.7, and WFYT025 12.33 at 1 DPA, WFB 10.97, CHT025 10.03, and WFYT025 5.37 at 10 DPA. Compared to WFB and CHT025, WFYT025 had greater leaf length, leaf width, and leaf area at 1 DPA and 10 DPA.

Heterosis of the middle parent (MPH) and Heterosis of the higher parent (HPH) were estimated for flag leaf Heterosis (Table 1). MPH showed a negative effect on leaf length around early rice and TWK around middle rice at 10 DPA. Except for MPH mentioned above, MPH values for all traits varied from 5.3 to 43.4%. Most of the phenotype parameters of WFYT025 show MPH, although a few traits have high parental Heterosis HPH.

### 2.2. RNA Sequencing of WFYT025 and Its Parents

A total of 1725.99 million reads were generated, with an average of 47.94 million reads per sample. The total number of valid reads is 1559.21 million, and the average valid read per sample is 43.31 million. In each case, the Q20 fraction was over 99%, and the Q30 base percentage was over 95% (Appendix A). Therefore, the quality of the data was very high and met the requirements for further analysis.

To confirm the accuracy and reproducibility of the RNA-Seq results, 12 genes were selected for qRT-PCR with the specific primers (Appendix A). The validation results for the 12 genes are shown in Appendix A. The qRT-PCR results were all in agreement with the RNA-Seq data. Thus, our transcriptome sequencing results were credible.

### 2.3. Analysis of Differentially Expressed Genes

To analyze gene expression at different stages under different environmental conditions, gene expression was mainly measured by FPKM (Fragments per Kilobase of exon model Per Million mapped reads) or RPKM (reads Per Kilobase of exon model Per Million mapped reads) (Figure 1A). In the leaves of CHT025 under the early rice environment, the number of DEGs (*p* < 0.05) at 1 DPA and 10 DPA totaled 3072, of which 1937 were up-regulated, and 1135 were down-regulated. At the same time, the number of DEGs in WFB leaves at 1 DPA and 10 DPA were 5039 in total, among which 3244 were up-regulated, and 1795 were downregulated. In WFYT025 leaves, the number of DEGs at 1 DPA and 10 DPA was 3757, of which 1937 were up-regulated, and 1135 were down-regulated (Figure 1B). In the environment of middle rice, the number of genes up-regulated in CHT025, WFB, and WFYT025 was 892, 1273, and 819, respectively, whereas the number of genes down-regulated in CHT025, WFB, and WFYT025 was 616, 1934, and 2196, respectively (Figure 1C).

### 2.4. Gene Ontology (GO) Annotation and KEGG Pathway Enrichment Analysis

GO enrichment analysis of these DEGs identified several significant biological processes at different stages and under different environmental conditions. In the environment of early rice, “chloroplast stroma” and “oxidation-reduction” processes were significantly enriched in CHT025 (Figure 2A). Meanwhile, “closed chloroplast”, “ATP binding”, “cytosol”, and “mitochondrion” were significantly enriched in WFB (Figure 2B). At the same time, “the cytosol”, “chloroplast”, and “oxidation-reduction” processes were significantly enriched in WFYT025 (Appendix A). In the environment of the middle rice, “protein serine/threonine kinase activity”, “protein phosphorylation”, “extracellular region”, and “plasma membrane” were significantly enriched in CHT025 (Appendix A). “Closed protein serine/threonine kinase activity”, “ATP binding”, “plasma membrane”, and “defense response” were significant enriched in WFB (Appendix A). At the same time, “ATP binding”, “integral component of the membrane”, “plasma membrane” and “protein serine/threonine kinase activity” were significantly enriched in WFYT025(Appendix A). This suggests that the substantial differences in flag leaf between 1 DPA and 10 DPA may be related to photosynthetic efficiency.

Functional enrichment analysis was performed for all these DEGs during rice seed development. A total of 131 signaling pathways were identified in 1502 DEGs (including the 1 DPA and 10 DPA of CHT025 in the early rice environment). KEGG enrichment analysis showed that the t20 most enriched pathways included “ubiquinone”, “other terpenoid-quinone biosynthesis”, “glycerolipid metabolism”, “taurine”, “hypotaurine metabolism” and “tryptophan metabolism,” etc. (Figure 3A). Meanwhile, 2663 DEGs in WFB were classified into 137 metabolic pathways, and the 20 most enriched metabolic pathways mainly focused on concentrated “starch”, “sucrose metabolism”, “fatty acid metabolism”, “nicotinate”, “nicotinamide metabolism”, “fatty acid elongation,” etc. (Figure 3B). In total, 1911 DEGs in WFYT025 were classified into 133 pathways, and the 20 most enriched pathways mainly focused on concentrated “ribosome”, “purine metabolism”, “nitrogen metabolism” and “ribosome biogenesis in eukaryotes,” etc. (Appendix A). In the environment of middle Rice, the pathways “plant–pathogen interaction”, “cutin”, “suberine”, “wax biosynthesis”, “brassinosteroid biosynthesis”, and “flavonoid biosynthesis” were enriched in CHT025 between 1 DPA and 10 DPA. In contrast, the pathways “plant-pathogen interaction”, “diterpenoid biosynthesis”, “phenylalanine metabolism”, and “flavonoid biosynthesis” were enriched in WFB. While “plant-pathogen interaction”, “amino sugar”, “nucleotide sugar metabolism”, “diterpenoid biosynthesis”, and “AGE-RAGE signaling” pathway in diabetic complications were enriched in WFYT025.

### 2.5. Identification of Gene Co-Expression Modules in the Flag Leaf of WFYT025

After obtaining the gene expression profile of the flag leaf, we further investigated the gene regulatory network. Analysis of the gene regulatory network revealed several large subnetworks representing interactions between genes with similar expression profiles, which were termed co-expression modules [19]. The co-expressed gene sets via WGCNA were identified, and a total of 5000 highly expressed genes were divided into 24 co-expression groups (Figure 4). We then analyzed the correlation between the gene expression profile of the co-expression module and the three phenotypic data (TGW; length; width) (Appendix A). Since the goal is to find genes related to grain development, we focused on the co-expression module with a high correlation with TGW (the color of the module is Midnight blue, and the gene expression profile in this module is moderately negatively correlated with the thousand-grain weight phenotype). We found the overall gene expression profile of the co-expression module Midnight Blue, which had the highest correlation with TGW, and the co-expression module contained a total of 106 genes (Appendix A). The GO enrichment analysis of these genes revealed several significant biological processes. Modules associated with TKW showed biological processes that are associated with “protein amino acid phosphorylation”, “protein modification process”, and so on (Figure 5). Overall, these modules represent the specific gene regulatory processes at each stage and are the indicators of operated programs of flag leaf development.

## 3. Discussion

In the plant life cycle, leaves play a significant role in grain yield, and flag leaf length has been recognized as an important factor that determines plant type for high-yield potential in rice [20]. However, most of the previous studies on rice yield focused on rice grains, and few focused on the effect of rice flag leaf on rice yield. In the current study, we investigated the relationship between transcriptional profiles and heterosis in super hybrid rice WFYT025 by RNA-Seq.

### 3.1. The Genetic Basis of Heterosis

A significant difference gene (SDG) was defined as the significant difference gene between 1 DPA and 10 DPA. Meanwhile, SDGs that existed in hybrid but not in parents were defined as SDG_hp_. We have been able to identify a number of SDG_hp_s underlying flag leaf between hybrid WFYT025 and paternal line CHT025, confirming the suggestion that heterosis is a polygenic phenomenon [21]. Under the environment of early rice, among the SDG_hp_s, 10.9% had a dominant effect, 41.81% had a partial dominant effect, 22.07% had an additive effect, and the remaining 25.22% had an over-dominant effect at 1 DPA. Meanwhile, at 10 DPA, among the SDG_hp_s, 12.28% had a dominant effect, 42.04% had a partial dominant effect, 18.52% had an additive effect, and the remaining 27.16% had an over-dominant effect. Thus, partial dominance was the major contributor to the flag leaf heterosis of WFYT025 (Figure 6)

However, at 1 DPA under the environment of middle rice, among the SDG_hp_s, 43.19% were partially dominant, 16.65% were dominant, 32.17% were over-dominant, and 7.99% had an additive effect. In contrast, at 10 DPA, 52.83% of the genes were super dominant, 26.54% partial dominant, 9.56% dominant, and 11.07% additive. These results might be due to high temperature, which was inconsistent with the results of other environments (Figure 7).

### 3.2. Transcription Factors May Underlie Heterosis

It is well known that the analysis of expression patterns of important transcription factors is helpful in understanding the heterosis in rice. In CHT025, we found 13 transcription factors of the WRKY family in SDG_hp_s under the environment of early rice and 16 genes of the WRKY family in SDG_hp_s of under the environment of middle rice. Transcription factors, including *WRKY3*, *WRKY68*, and *WRKY77*, were found in both environments. Recently, studies have shown that WRKY family genes are involved in the process of seed growth and matching in Arabidopsis and rice [22]. However, the expression level of *WRKY68* in *WFYT025* and its parents at 10 DPA was significantly higher than that of 1 DPA. The expression level of *WRKY68* in WFYT025 was significantly higher than that of its parents. In the two stages, we found nine identical transcription factors. Except for five reported TFs, the other four TFs might play an important role in grain number and photosynthesis heterosis.

### 3.3. The Key Metabolic Pathways of Heterosis

In order to eliminate the impact of the environment, we studied the metabolic pathway with the same SDG_hp_ (SSDG_hp_) in two environments. There were 312 SSDG_hp_s in total. These SSDG_hp_s were mainly concentrated in the phosphorus metallic process, phosphorylation, plasma membrane, etc. During senescence in annual crop plants, P (phosphorus) is mobilized from leaves and other vegetative tissues and translocated to seed development, which is a strong P sink during the reproductive growth phase [23,24,25]. Previous studies have shown that seed development is a strong sink for P, and the remobilization of P from vegetative tissues results in insufficient P for other processes necessary for continued growth, such as photosynthesis. In rice (*Oryzasativa* L.), P is predominantly mobilized from leaves during the later stages of grain filling [26,27]. This might be the reason why the filling speed of rice was faster from 1 DPA to 10 DPA after flowering than from 15 DPA to 21 DPA.

## 4. Material and Methods

### 4.1. Plant Materials and Growth Conditions

The hybrid WFYT025, along with its parental lines Changhui T025 (CHT025) and Wufeng B (WFB), were planted in the experimental field of Jiangxi Agricultural University, Nanchang (28°450′ N, 115°500′ E) in 2019. The sowing time of early rice was on 17 March, and the sowing time of middle rice was on 20 May. WFYT025 is a super-hybrid rice combination derived from the cross between female parent WFB and male parent CHT025. Each plot consisted of 40 rows, with each row having 10 plants at a 20 cm planting space. Crop management followed normal procedures for rice. The three lines were chosen in this study to measure phenotypic traits and conduct transcriptome analyses. We sampled the grains and flag leaf on the first day, the tenth day, the fifteenth day, and the twenty-first day after the anthesis of rice. By measuring the TGW of the grains, we finally selected the flag leaf at 1DPA and 10 DPA for sequencing. The flag leaf was collected and stored at −80 °C for RNA-Seq analysis, and each sample had at least three biological replications to minimize systematic errors.

### 4.2. RNA Extraction, Sequencing, De Novo Assembly and Statistical Methods

Total RNA was extracted from rice leaf using Trizol reagent (Invitrogen, Carlsbad, CA, USA) and purified using an RNeasy Plant Mini Kit (Qiagen, Valencia, CA, USA) according to the manufacturer’s instructions. The integrity of RNA was verified by RNase-free agarose gel electrophoresis, and the concentration was measured with a 2100 Bioanalyzer (Agilent Technologies, Santa Clara, CA, USA). cDNA library was constructed using the mRNA fragments as templates and sequenced on the Illumina HiSeq™ 4000 platform (Illumina Inc., San Diego, CA, USA). Adapter sequences were removed from the raw reads. It follows that low-quality reads with 50% bases with quality scores of five or lower and/or over 10% bases unknown were removed to gain more reliable results. The raw data generated by sequencing were preprocessed by following Lianchuan Biological cutadapt procedure to filter out unqualified sequences to obtain valid data (clean data) before proceeding to the next step of the analysis. Raw data were cleaned through the following processing steps: (1) removing the reads with connectors (Adaptor); (2) removing the reads containing N (N indicates that base information cannot be determined) with a percentage greater than 5%; (3) removing low quality reads (the number of bases with quality value Q ≤ 10 accounts for more than 20% of the whole read); (4) counting raw sequencing volume, effective sequencing volume, Q20, Q30, GC content, and comprehensive analysis.

### 4.3. Pathway Enrichment Analysis

KEGG (Analysis of Kyoto Encyclopedia of Genes and Genomes) is the major public pathway-related database. Pathway enrichment analysis identified significantly enriched metabolic pathways or signal transduction pathways in DEGs (different expression genes) compared with the whole genome background. Differential enrichment analyses, including Gene Ontology and KEGG pathway enrichment analysis, enrichment analysis using the cluster profile R software package (edgeR).Significantly enriched pathways in DEGs compared to the genome background were defined by a hypergeometric test. The calculated *p*-value was gone through FDR Correction, taking FDR ≤ 0.05 as a threshold. Pathways meeting this condition were defined as significantly enriched pathways in DEGs.

### 4.4. GO Enrichment Analysis

Gene Ontology (GO) is an international standardized gene functional classification system that offers a dynamic-updated controlled vocabulary and a strictly defined concept to comprehensively describe the properties of genes and their products in any organism. GO has three ontologies: molecular function, cellular component, and biological process. The basic unit of GO is GO-term. Each GO term belongs to a type of ontology.

GO enrichment analysis provides all GO terms that are significantly enriched in DEGs compared to the genome background, filtering the DEGs that correspond to biological functions. GO enrichment analysis was performed using the OmicShare tools, a free online platform for data analysis (www.omicshare.com/tools accessed on 1 May 2023). Firstly all DEGs were mapped to GO terms in the Gene Ontology database (http://www.geneontology.org/ accessed on 1 May 2023), gene numbers were calculated for every term, and significantly enriched GO terms in DEGs compared to the genome background were defined by hypergeometric test. The calculated *p*-value was gone through FDR Correction, taking FDR ≤ 0.05 as a threshold. GO terms meeting this condition were defined as significantly enriched GO terms in DEGs. This analysis was able to recognize the main biological functions that DEGs exercise.

### 4.5. Quantitative Real-Time PCR (qRT-PCR) Validation

To validate the RNA-seq results, different expression patterns of several genes were confirmed by quantitative real-time RT-PCR (qRT-PCR). For qRT-PCR, 1 µg of total RNA was used to synthesized cDNA using PrimeScript^TM^ RT reagent Kit (Perfect Real Time) (Dalian, China, TaKaRa). The qRT-PCR was carried out using SYBR^®^ Premix Ex Taq II (Tli RNaseH Plus; TAKARA BIO Inc., Shiga, Japan) and determined in LightCycler 480 (Roche, Basel, Switzerland) according to the manufacturer’s instructions. The qRT-PCR reactions were amplified for 95 °C for 30 s, followed by 40 cycles of 95 °C for 5 s, 55 °C for 30 s and 72 °C for 30 s. All reactions were performed with three independent biological replicates for each sample, and three technical replicates for each biological replicate were analyzed. The relative gene expression was calculated by the software of ABI7500 Real-Time PCR System using the 2^−∆∆Ct^ method. The detection of the threshold cycle for each reaction was normalized against the expression level of the rice Actin1 gene with the primer sequences 5′-TGGCATCTCTCAGCACATT CC-3′ and 5′-TGCACAATGGATGGGTCAGA-3′.

### 4.6. The Mode of Inheritance Analysis

For statistical analysis, the analysis of variance (ANOVA) was usually by the model: y = u + (GA) + (GD) + (SR) + e, where y is the acquired gene expression, u is the overall mean, GA is the additive effect, GD is the dominant effect, SR is the replication effect, and e is the residual error (Chen L et al., 2018). Hp = [d]/[a], referred to as the dominance ratio or potency (where [a] and [d] represent GA and GD, respectively). Considering gene expression levels as quantitative traits, we adopted traditional quantitative genetic parameters, such as composite additive effect [a] and composite dominance effect [d], to estimate our expression profile. DGHP were classified according to the dominance ratio Hp (=[d]/[a]), based on 99.8% confidence intervals constructed for [d] − [a] ([d] > 0) and [d] + [a] ([d] < 0). According to the value of Hp (=[d]/[a]), we considered that these genes belonged to partial dominance (−0.8 < Hp ≤ −0.2 or 0.2 < Hp ≤ 0.8), over-dominance (Hp ≤ −1.2 or Hp > 1.2), dominance (−1.2 < Hp ≤ −0.8 or 0.8 < Hp ≤ 1.2), and additive effect (−0.2 < Hp ≤ 0.2) [28,29].

### 4.7. Normalization of Gene Expression Levels and Identification of Differentially Expressed Genes

Sequencing reads were mapped to the reference sequences. The expression level of each gene was measured by fragments per kilobase of exon model per Million mapped reads (FPKM). To determine the time-dependent transcriptional differences between early rice and middle rice, the differential expression genes (DEGs) at 1 DPA and 10 DPA were determined by comparing the expression levels. To correct for multiple testing, the false discovery rate (FDR) was calculated to adjust the threshold of the *p*-value [30]. The standard of different expressions between compare is a minimal 2-fold difference in expression (|log2 Ratio| ≥ 1) and an FDR ≤ 0.005 [31].

### 4.8. Weighted Gene Co-Expression Network Analysis (WGCNA)

The gene co-expression network analysis used the raw expression counts (RAW counts) of all Rice reference genes in the RNA-seq data of the 36 leaf groups obtained from the previous analysis. Firstly, the genes with low expression were eliminated. The elimination standard was that if the genes were not expressed in more than 80% of the samples, the genes with low expression were considered. Then, the median absolute deviation was selected from the genes with high expression(mad) 5000 genes of the maximum value for total express network analysis using WGCNA (https://horvath.genetics.ucla.edu/html/CoexpressionNetwork/Rpackages/WGCNA/ accessed on 1 May 2023). In the process of WGCNA analysis, 20 soft power was selected. GO enrichment analysis through the agriGOv2 (http://systemsbiology.cau.edu.cn/agriGOv2/ accessed on 1 May 2023), its parameters as follows: Statistical Test Method as HyperGeometry, Multi_ Test Adjustment Method as Hochberg FDR, Significance level as “0.05” and a Minimum number of mapping entries as “5”.

## 5. Conclusions

The flag leaf of WFYT025 and its two parents were sampled for high-throughput transcriptome sequencing in identifying genes related to leaf photosynthesis or transpiration and the development of seeds. We selected the flag of hybrid rice WFYT025 and its parents for transcriptome sequencing, the 1 DPA and 10 DPA in the environment of early rice and middle rice, respectively. Several major sub-networks were found to represent interactions among genes with similar expression profiles via gene regulatory network analysis. The co-expressed gene sets via weighted gene co-expression network analysis (WGCNA) were identified, and a total of 5000 highly expressed genes were divided into 24 co-expression groups. In the two stages, we found nine identical transcription factors. Except for five reported TFs, the other four TFs may play an important role in grain number and photosynthesis heterosis.

## Figures and Tables

**Figure 1 plants-12-02496-f001:**
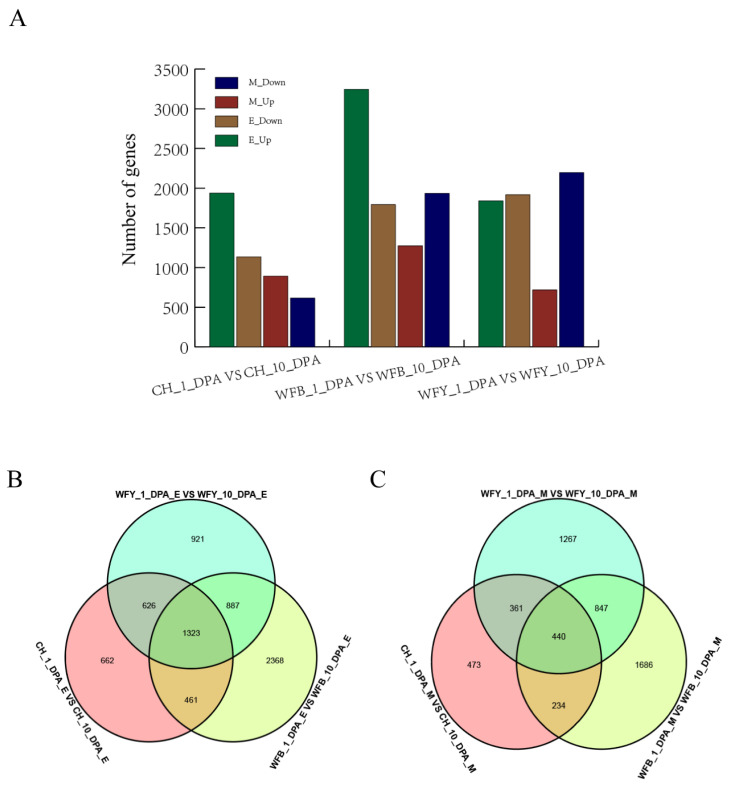
DEGs in super hybrid WFYT025 combination. (**A**) Number of DEGs between the hybrid and its parents in different stages under different environments. (**B**) Venn diagram of DEGs between the hybrid and its parents under the early rice environment (**C**) Venn diagram of DEGs between the hybrid and its parents under the middle rice environment. CH, WFY and WFB represent CHT025, WFYT025 and WFB, respectively. E and M represent early and middle rice environments.

**Figure 2 plants-12-02496-f002:**
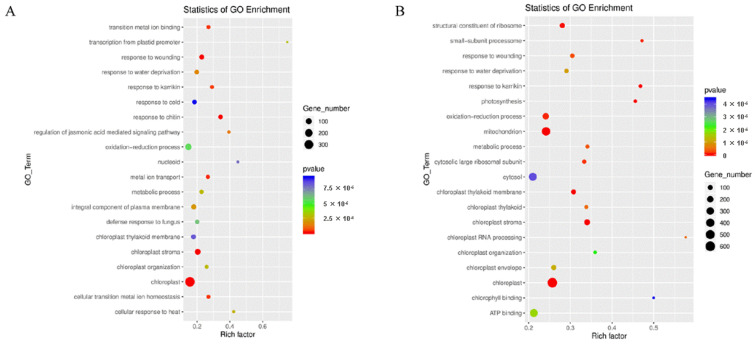
Gene ontology (GO) enrichment analysis of DEGs (**A**) GO enrichment analysis of DEGs at 1DPA and at 10 DPA under the environment of early rice in CHT025. (**B**) GO enrichment analysis of DEGs at 1DPA and at 10 DPA under the environment of early rice in WFB.

**Figure 3 plants-12-02496-f003:**
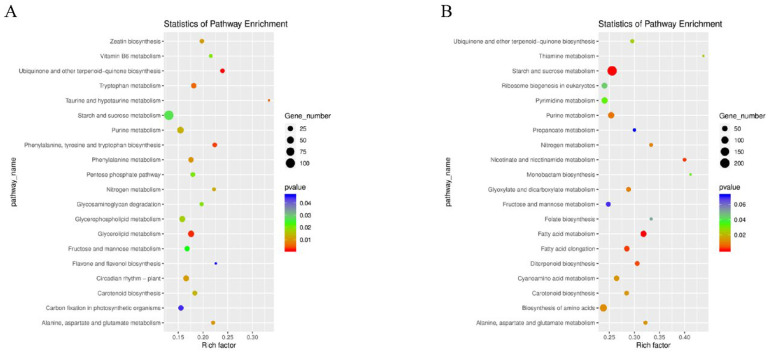
KEGG pathway assignments of DEGs. (**A**) KEGG analysis of DEGs between CHT025 in the environment of early rice. (**B**) KEGG analysis of DEGs between WFB in the environment of early rice. (**A**,**B**) showed the top 20 most represented categories and the number of transcripts predicted to belong to each category.

**Figure 4 plants-12-02496-f004:**
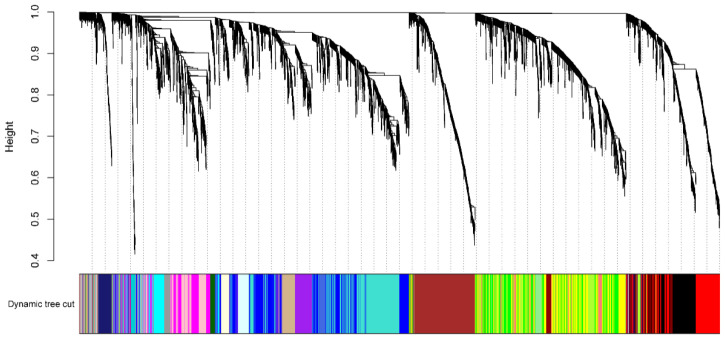
Analysis results of 5000 genes in co-expression module. The top half of the graph is divided into a clustering tree of 5000 genes based on their expression profiles into 36 sets of RNA Seq data, with each branch representing one gene.The bottom half of the graph is divided into co-expression modules corresponding to the 5000 genes, and the same co-expression modules are represented by the same color. The analysis resulted in 24 co-expression modules.

**Figure 5 plants-12-02496-f005:**
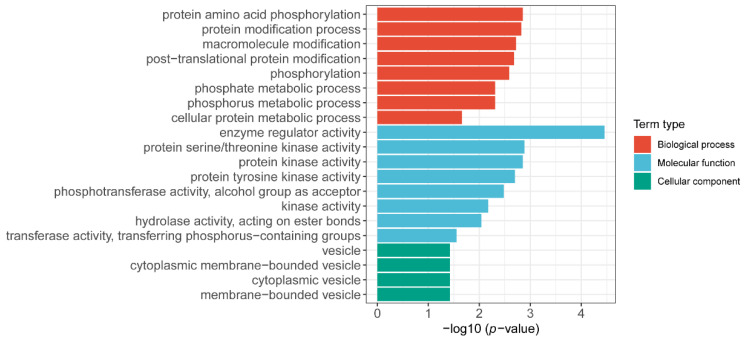
Go enrichment analysis of midlightblue module about 106 genes in the co-expression module showed statistically significant GO term (FDR < 0.05). The x-axis indicates the level of statistical significance, and the y-axis indicates the significantly enriched GO term.

**Figure 6 plants-12-02496-f006:**
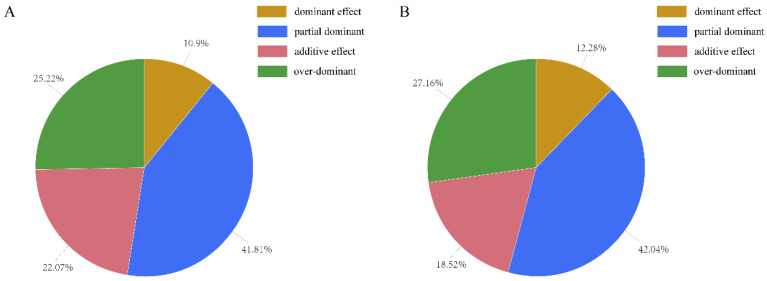
Breakdown of the SDG_hp_s according to the dominance ratio Hp under the environment of early rice. Depending on the principle of Hp = [d]/[a], Hp was classified as either positive or negative. (**A**) Breakdown of the SDG_hp_s according to the dominance ratio Hp On the 1st day after DAA. (**B**) Breakdown of the SDG_hp_s according to the dominance ratio Hp On the 10th day after DAA.

**Figure 7 plants-12-02496-f007:**
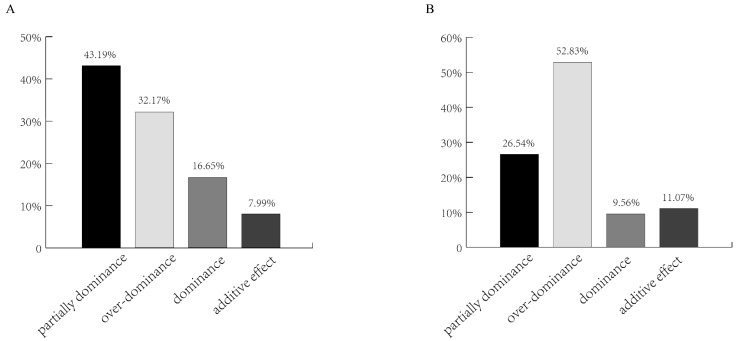
Breakdown of the SDG_hp_s according to the dominance ratio Hp under the environment of middle rice. Depending on the principle of Hp = [d]/[a], Hp was classified as either positive or negative. (**A**) Breakdown of the SDG_hp_s according to the dominance ratio Hp on the 1st day after DAA. (**B**) Breakdown of the SDG_hp_s according to the dominance ratio Hp On the 10th day after DAA.

**Table 1 plants-12-02496-t001:** Phenotypic analysis of WFYT025 and its parents of early rice and middle rice.

Environment	Traits	WFB	CHT025	WFYT025	MPH (%)	HPH (%)

early rice	1D leaf length (cm)	28.17 ± 1.485	25.2 ± 1.92	30.4 ± 2.32	13.9 *	7.9
	1D leaf width (cm)	2.02 ± 0.107	2.18 ± 0.33	2.38 ± 0.30	13.3 *	9.1 *
	1D leaf area (cm^2^)	42.68	41.2	54.26	29.3 **	27.1 **
	1D 1000 spikelet weight (g)	6.7	5.68	7.3	17.9 **	8.9

	10D leaf length (cm)	31.8 ± 3.585	30.15 ± 1.80	30.18 ± 1.29	−2.6	−5
	10D leaf width (cm)	2.07 ± 0.080	2.33 ± 0.22	2.35 ± 0.28	6.8	0.8
	10D leaf area (cm^2^)	49.37	52.69	53.19	4.2	0.1
	10D TGW (g)	24.86	14.77	20.42	2.81	−17.8 **

middle rice	1D leaf length (cm)	27	38.5	44.1	34.7 **	14.5
	1D leaf width (cm)	1.93	2	2.1	6.9	5
	1D leaf area (cm^2^)	39.08	57.75	69.46	43.4 **	20.2 **
	1D 1000 spikelet weight (g)	5.03	4.25	4.84	4.3	−3.7

	10D leaf length (cm)	31.37	40.78	43.68	21.1 **	7.1
	10D leaf width (cm)	2.11	2.22	2.28	5.3	2.7
	10D leaf area (cm^2^)	49.64	67.9	74.69	27.1 **	10
	10D TGW (g)	29.06	14.64	21.1	−3.4	−27.3 **

** Significant difference with *p* < 0.01, * Significant difference with *p* < 0.05.

## Data Availability

Data are available upon request.

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
