# Peer review of "Genetic Analysis for the Flag Leaf Heterosis of a Super-Hybrid Rice WFYT025 Combination Using RNA-Seq"

_plants, 2023, doi:10.3390/plants12132496_

Round 1

Reviewer 1 Report (Previous Reviewer 2)

This article takes the super rice hybrid variety WFYT025 and its parents as the research objects, and the transcript differences between the three and the differences in heterosis under early and middle rice planting conditions were compared. The key pathways and genes of heterosis were explained. The research is relatively complete, and the experimental design is complete, which has reference significance. The writing of this article is authentic, fluent, and meets the publishing requirements.

However, there are some minor problems, like language grammar issues, need to be carefully revised .So it is suggested to revise and publish.

Author Response

Dear reviewer:

Thank you for your comments. We have carefully checked and revised the manuscript.

Reviewer 2 Report (New Reviewer)

In the submitted manuscript, Cheng et al. provided the genetic basis of heterosis for the flag leaf in India super hybrid rice combination WFYT025 using RNA-seq strategy, which provides resources for understanding heterosis during the flag leaf development . Here are my detailed comments.

1. The authors should use the Microsoft Word template of 2023, https://www.mdpi.com/journal/plants/instructions.

2. English writing should be carefully revised by a native speaker.

3. The authors should consistent formatting of references.

4. Specific statistical methods should be provided for supplementary Table 1. And the authors should annotate early rice and middle rice in Table 1 legend.

5. The specific information of the materials needs to be supplemented in Table S1 legend

6. Supplementary Fig. S1, D-E should change to D-F.

7. Please provide more specific information for Fig. 4.

8. Some Figure legend in the draft seems too simple. The authors should provide detailed information.

English writing should be carefully revised by a native speaker.

Author Response

Comments and Suggestions for Authors

In the submitted manuscript, Cheng et al. provided the genetic basis of heterosis for the flag leaf in India super hybrid rice combination WFYT025 using RNA-seq strategy, which provides resources for understanding heterosis during the flag leaf development. Here are my detailed comments.

R: Thank you for your comments, we have checked the manuscript and revised it according to the comments.

The authors should use the Microsoft Word template of 2023, https://www.mdpi.com/journal/plants/instructions.

R1: Okay, it has been revised.

English writing should be carefully revised by a native speaker.

R2: A native speaker has revised English writing carefully.

The authors should consistent formatting of references.

R3: Thanks for the comment, we have revise it.

Specific statistical methods should be provided for supplementary Table 1. And the authors should annotate early rice and middle rice in Table 1 legend.

R4: Thanks for the comment, we have supplemented specific statistical methods in the materials and method, and annotate early rice and middle rice in Table 1 legend.

The specific information of the materials needs to be supplemented in Table S1 legend

R5: Thanks for the comment, we have supplemented the specific information of the materials in Table S1 legend.

Supplementary Fig. S1, ‘D-E’ should change to ‘D-F’.

R6: Thanks for the comment, we have revise it.

Please provide more specific information for Fig. 4.

R7: We have explained Fig. 4 in detail

Some Figure legend in the draft seems too simple. The authors should provide detailed information.

R8: Thanks for the comment ,we have explained Figure legend in detail.

Round 2

Reviewer 2 Report (New Reviewer)

The author has answered all the comments I suggested for the first time, I think the MS can be accepted.

This manuscript is a resubmission of an earlier submission. The following is a list of the peer review reports and author responses from that submission.

Round 1

Reviewer 1 Report

The aim of this work is to investigate heterosis in the flag leaf of a high-yielding hybrid as seen by RNA-seq gene expression.

The authors used the hybrid and its two parents, who planted them in two separate environments. The effort made to collect data from plants under real growing conditions is noteworthy, but here the authors did not pay attention to the environmental variability that conditions gene expression. The environment can change between sampling dates and even more so between sites. Authors should collect weather data and use these to evaluate the results obtained. For example, the hybrid has more downregulated genes in the middle environment and more upregulated genes in the early environment.

Other questions that authors ought to respond to are:

-In Materials and Methods:

- the authors say that sowing was carried out in two locations (Nanchang and Hainan) with different variations in day duration conditions, but no sowing date is indicated.

- What specific plots are the authors referring to? A single genotype? What number of replicates were performed? What statistical design was used?

Why is TGW measurement crucial when choosing the flag leaf collection date for RNA-seq?.

- No information is given regarding how the phenotypic data was obtained.

- In the Results, the table containing phenotypic data is missing.

- in 2.1 authors say that at 1 DPA WFYTO25 had a higher TGW regarding parentals, but no grain exists at 1 DPA, the weight is mainly spikelets.

- In supplementary figures 4 and 5 Thousand grain weight is abbreviated as TKW instead of TGW as in 2.1.

Reviewer 2 Report

This article takes the super rice hybrid variety WFYT025 and its parents as the research objects, and the transcript differences between the three and the differences in heterosis under early and middle rice planting conditions were compared. The key pathways and genes of heterosis were explained. The research is relatively complete, and the experimental design is complete, which has reference significance. The writing of this article is authentic, fluent, and meets the publishing requirements.

1. In 4.1. Plant materials and growth conditions, the author wrote “Nanchang province”, this should be revised.

2. There are differences in the formatting of many paragraphs in the manuscript, please unify and improve them.

3. The format of references 6 and 31 is inconsistent with others, please check for updates.

4. Please indicate the corresponding content of the first letter abbreviation in the article, such as DHA.

5. What are the screening criteria for SDG? Please specify.

Minor editing of English language required